# Teaching hospitals and their influence on survival after valve replacement procedures: A retrospective cohort study using inverse probability of treatment weighting (IPTW)

**Kevin Maldonado-Cañón**[1]*, **Giancarlo Buitrago**[1,2], **Germán Molina**[3], **Francisco Mauricio Rincón Tello**[4,5], **Javier Maldonado-Escalante**[3,4]

1 Facultad de Medicina, Instituto de Investigaciones Clínicas, Universidad Nacional de Colombia, Bogotá, Colombia, 2 Hospital Universitario Nacional de Colombia, Bogotá, Colombia, 3 Department of Cardiovascular Surgery, Clínica Universitaria Colombia, Bogota, Colombia, 4 Department of Cardiovascular Surgery, Hospital Universitario Fundación Santa Fe de Bogotá, Bogota, Colombia, 5 Department of Cardiovascular Surgery, Clínica Los Nogales, Bogotá, Colombia

* kmaldonadoc@unal.edu.co

**Data Availability Statement:** Data cannot be shared publicly because of a confidentiality agreement between Universidad Nacional de

## Abstract

### Background

The effect of teaching hospital status on cardiovascular surgery has been of common interest in recent decades, yet its magnitude on heart valve replacement is still a matter of debate. Given the ethical and practical unfeasibility of randomly assigning a patient to such an exposure, we use the inverse probability of treatment weighting (IPTW) to assess this marginal effect on the survival of Colombian patients who underwent a first heart valve replacement between 2016 and 2019.

### Methods

A retrospective cohort study was conducted based on administrative records. The time-to-death event and cumulative incidences of death, readmission, and reoperation are presented as outcomes. An artificial sample is configured through IPTW, adjusting for sociodemographic variables, comorbidities, technique, and intervention weight.

### Results

Of a sample of 3,517 patients, 1,051 (29.9%) were operated on in a teaching hospital. The median age was 65.0 (18.1–91.5), 38.5% of patients were ≤60, and 6.9% were ≥80. The cumulative incidences of death at 30, 90 days, and one year were 5.9%, 8%, and 10.9%, respectively. Furthermore, 23.5% of the patients were readmitted within 90 days and 3.6% underwent reinterventション within one year. The odds of 30-day mortality are lower for patients operated in a teaching hospital (OR 0.51; 95% CI 0.29–0.92); however, no effect on survival was identified in terms of time-to-event of death (HR 1.07; 95%CI 0.78–1.46).

Colombia and the Ministry of Health of Colombia. Data are available from the integrated information system of the Ministry of Health of Colombia (https://www.sispro.gov.co/Pages/Home.aspx) for researchers who meet the criteria for access to confidential data."

**Funding:** The author(s) received no specific funding for this work.

**Competing interests:** The authors have declared that no competing interests exist.

## Conclusions

After IPTW, the odds of 30-day mortality are lower for patients operated in a teaching hospital. There was no effect on survival, 90-day or one-year mortality, 90-day readmission, or one-year reintervention. Together, we offer an opening for investigating an exposure that has yet to be explored in Latin America with potential value to understand teaching hospitals as the essential nature of reality of an academic-clinical synergy.

## Introduction

The effect of teaching hospital status on cardiovascular surgery has been of common interest in recent decades [1]. Shortly before the 1990s, teaching hospitals began to be conceived as differential institutions with a wide reputation for providing high-quality clinical care and reflecting positive opinions in the collective imagination of the population [2]. However, our knowledge of the magnitude of this phenomenon on heart valve replacement is limited; several studies suggest that it has a favorable effect [3–6], while other studies do not show an effect [7–10].

Although the definitions for this "teaching hospital exposure" vary, they are built upon three main axes: 1. State regulation (i.e., certification before a national medical or governmental association), 2. Academic cooperation, and 3. Clinical training [2,11]. They are acknowledged for having cutting-edge technologies intertwined with an ideal of continuous improvement through clinical research [12] and -from a societal perspective- for providing care for vulnerable folks and minorities with unlike coverage and comprehensiveness [13].

Naturally, multiple hypotheses have been tested, posing the question of whether these differences, these attributes, and these apparent advantages of a teaching hospital affect clinical outcomes. Once adjusted for experience, volume, clinical complexity, and baseline risk, there appears to be no evident significant difference in overall survival; however, it depends on the nature of the specific clinical condition and the outcomes evaluated [14]. The lack of consistency and generalizability for heart valve surgery may be attributable to the intrinsic variability of each country's multidimensional healthcare context, understood as an utterly particular microcosm.

Considering the knowledge gap in Latin America in this regard, and the ethical and practical unfeasibility that would demand randomly assigning a patient to care in a teaching or nonteaching hospital in a clinical trial, we used the inverse probability of treatment weighting (IPTW) to assess the marginal effect of teaching hospital status on the survival of patients affiliated with the contributory regime in Colombia who underwent a first heart-valve replacement between 2016 and 2019.

## Materials and methods

### Source of information, patients, and exposure

A retrospective cohort study was conducted based on administrative records. We used the Capitation Sufficiency Database from the Colombian Ministry of Health as the primary source of information. It contains deidentified patient-level data on the consumption of healthcare services, demographic information, and associated ICD-10 codes. Comorbidities were identified through ICD-10 codes, and the Charlson Comorbidity Index (CCI) was calculated using the original score validated in our database [15].

The Colombian health system comprises mainly two regimens: the contributory regime (patients with formal employment) and the subsidized regime (patients without formal employment in low-resource settings). The available information includes claims of services provided by insurers (Health Promoter Enterprises; EPS in Spanish), comprising approximately 80% of the contributory regime, covering 22.19 million Colombians in 2016 (48% of the total population). Additionally, the date of death was obtained from the national database of death certificates. Both databases have been previously utilized, which supports their suitability [15].

Patients were ≥18 and underwent an isolated or combined valve replacement procedure between January 1, 2016, and December 31, 2019 (index procedure). All patients with previous cardiac surgery identified from January 2011 to the index date of the procedure were excluded. Due to the lack of a standardized definition in our country, the "Teaching Hospital" status was defined as either being officially accredited and certified by the Ministries of Education and Health as a teaching hospital or, despite not meeting the previous definition, being a hospital that partners with a medical school to provide training to cardiac surgery fellows.

## Outcomes and analysis

Our primary outcome was time-to-event of death. As secondary outcomes, we considered the cumulative incidence of death (mortality) at 30 days, 90 days, and one year, readmission at 90 days, and reintervention at one year; Poisson distribution was used to calculate the 95% confidence intervals (CI). In addition to the variables originally contained in the database, we included as predictors the intervention weight (i.e., isolated or combined procedures), the intervention technique (surgical (i.e., open surgery), minimally invasive, or transcatheter), and the categorized CCI (none (0), mild (1–2), moderate (3–4), and severe (≥5)) [16]. We validated the normality assumptions using graphic methods and described the variables accordingly. Secondary outcomes were estimated for the entire sample and categorized by age group, sex, CCI categories, region (S1 Table), intervention weight, intervention technique, and teaching hospital status.

IPTW using the propensity score was employed to estimate the causal effect of our exposure by assembling an artificial weighted balanced sample. As proposed by Austin et al. [17], variables reported in the literature to confer a prognostic value and that could act as confounders were included in the model specification.

IPTW with restriction was implemented for patients whose propensity score lay in the interval 0.1 to 0.9 [18], assuming a strong exposure selection process (since most of the teaching hospitals are located in Bogota (n = 6, 46.1%) and the Central region (n = 5, 38.5%), and some insurers have a direct agreement with certain teaching hospitals). With the weighted sample, we estimated the relative average effect of the treatment using a univariate Cox proportional hazards model; the Kaplan-Meier survival curves were also calculated to estimate the absolute effect of exposure.

Furthermore, univariate logistic regression models were fitted to evaluate the relative effect on secondary outcomes by estimating odds ratios (OR). Robust standard errors were calculated considering the weighting process. Positive and negative results were reported and those in which a statistical significance of the p-value <0.05 was not reached, considering their eventual clinical significance. All analyzes were completed using the statistical language R (version 4.0.3; R Core Team, 2020) (S1 Text). This study was approved by the IRB of the School of Medicine of the National University of Colombia. Individual informed consent was not obtained due to the nature of the data.

## Results

### Descriptive analysis

We included data from a total of 3,517 patients in the analysis. 38.5% were ≤60, 28.7% were between 60–70, 25.9% were between 70–80, and 6.9% were ≥80 years at the time of the index procedure. Additionally, 29.9% of the patients were operated on in one of the 13 hospitals identified as teaching hospitals. Table 1 shows detailed information on baseline characteristics by exposure status.

Regarding comorbidities, congestive heart failure (n = 1,001, 28.5%) was more frequent, followed by diabetes mellitus (n = 808, 23%), chronic obstructive pulmonary disease (COPD) (n = 769, 21.9%), acute myocardial infarction (n = 742, 21.1%), kidney disease (n = 705, 20%), cancer (including leukemia and lymphoma) (n = 485, 13.8%), and peripheral vascular disease (n = 397, 11.3%). With frequencies less than 10%, there were cerebrovascular events (n = 284, 8.1%), connective tissue diseases (n = 264, 7.5%), diabetes complications (n = 123, 3.5%),

**Table 1. Sociodemographic and clinical characteristics.**

| | TH (N = 1,051) | Non-TH (N = 2,466) | Total (N = 3,517) | p-value |
|---|---|---|---|---|
| **Age in years** | | | | |
| Median (range) | 65.8 (18.1, 91.5) | 64.6 (18.7, 89.7) | 65.0 (18.1, 91.5) | 0.028[1] |
| **Sex** | | | | 0.032[3] |
| Male | 605 (57.6%) | 1,515 (61.4%) | 2,120 (60.3%) | |
| **CCI** | | | | |
| Median (range) | 1.0 (0.0, 15.0) | 2.0 (0.0, 16.0) | 2.0 (0.0, 16.0) | 0.053[1] |
| Mean (SD) | 2.0 (2.2) | 2.2 (2.5) | 2.2 (2.4) | 0.023[2] |
| **CCI—Categories** | | | | 0.254[3] |
| None (0) | 280 (26.6%) | 617 (25.0%) | 280 (26.6%) | |
| Mild (1–2) | 449 (42.7%) | 1042 (42.3%) | 449 (42.7%) | |
| Moderate (3–4) | 205 (19.5%) | 474 (19.2%) | 205 (19.5%) | |
| Severe (≥5) | 117 (11.1%) | 333 (13.5%) | 117 (11.1%) | |
| **Weight of procedure** | | | | 0.114[3] |
| Isolated valve procedure | 834 (79.4%) | 1,865 (75.6%) | 2,699 (76.7%) | |
| Double valve procedure | 19 (1.8%) | 45 (1.8%) | 64 (1.8%) | |
| Isolated valve + 1 procedure | 177 (16.8%) | 503 (20.4%) | 680 (19.3%) | |
| Double valve + ≥2 procedures | 3 (0.3%) | 14 (0.6%) | 17 (0.5%) | |
| Isolated valve + ≥2 procedures | 18 (1.7%) | 39 (1.6%) | 57 (1.6%) | |
| **Year of Surgery** | | | | < 0.001[3] |
| 2016[a] | 1 (0.1%) | 74 (3.0%) | 75 (2.1%) | |
| 2017 | 299 (28.5%) | 744 (30.2%) | 1,043 (29.7%) | |
| 2018 | 478 (45.5%) | 880 (35.7%) | 1,358 (38.6%) | |
| 2019 | 273 (26.0%) | 768 (31.1%) | 1,041 (29.6%) | |
| **Region** | | | | < 0.001[3] |
| Bogota | 718 (68.3%) | 635 (25.8%) | 1,353 (38.5%) | |
| Central | 297 (28.3%) | 754 (30.6%) | 1,051 (29.9%) | |
| Other* | 36 (3.4%) | 1,077 (43.7%) | 1,113 (31.6%) | |

[1.] Wilcoxon rank sum test

[2.] Linear Model ANOVA

[3.] Pearson's Chi-squared test. CCI: Charlson Comorbidity Index; TH: Teaching Hospital; a: The codes used to identify the procedures entered into force until the end of 2016, which explains the low prevalence of the 2016 procedures. Other* (region): Atlantic, Eastern, and Pacific.

human immunodeficiency virus (HIV) (n = 58, 1.6%), peptic ulcer (n = 45, 1.3%), metastases (n = 40, 1.1%), dementia (n = 34, 1%), liver disease (n = 17, 0.5%), hemiplegia (n = 14, 0.4%) and severe liver disease (n = 5, 0.1%).

The most frequent procedure was isolated valve replacement (n = 2,699, 76.7%), followed by a combination of an isolated valve replacement + another single procedure (n = 680, 19,3%), double valve replacement (n = 64, 1.8%), a combination of an isolated valve replacement + ≥2 other procedures (n = 57, 1.6%), and a combination of double valve replacement + ≥1 other procedure(s) (n = 17, 0.5%). Detailed information on individual procedures can be found in S2 Table.

Table 2 shows the 30-day, 90-day, and one-year mortality categorized by our variables of interest. Of the entire cohort, 209 (5.9%) deaths were reported at 30 days, 280 (8%) at 90 days, and 383 (10.9%) at one year. The incidence rate of mortality was 6.6 deaths per 100 patient-years

**Table 2. Cumulative incidence of death at 30 days, 90 days, and one year.**

| | 30-day status | | 90-day status | | One-year status | |
|---|---|---|---|---|---|---|
| | Dead (n = 209) | Alive (n = 3,308) | Dead (n = 280) | Alive (n = 3,237) | Dead (n = 383) | Alive (n = 3,134) |
| **Age Group** | | | | | | |
| ≤ 60 | 53 (3.9%) | 1,301 (96.1%) | 67 (4.9%) | 1,287 (95.1%) | 87 (6.4%) | 1,267 (93.6%) |
| 60–70 | 68 (6.7%) | 942 (93.3%) | 89 (8.8%) | 921 (91.2%) | 119 (11.8%) | 891 (88.2%) |
| 70–80 | 70 (7.7%) | 841 (92.3%) | 98 (10.8%) | 813 (89.2%) | 132 (14.5%) | 779 (85.5%) |
| ≥ 80 | 18 (7.4%) | 224 (92.6%) | 26 (10.7%) | 216 (89.3%) | 45 (18.6%) | 197 (81.4%) |
| **Sex** | | | | | | |
| Male | 120 (5.7%) | 2,000 (94.3%) | 162 (7.6%) | 1,958 (92.4%) | 228 (10.8%) | 1,892 (89.2%) |
| Female | 89 (6.4%) | 1,308 (93.6%) | 118 (8.4%) | 1,279 (91.6%) | 155 (11.1%) | 1,242 (88.9%) |
| **CCI—Categories** | | | | | | |
| None (0) | 28 (3.1%) | 869 (96.9%) | 39 (4.3%) | 858 (95.7%) | 55 (6.1%) | 842 (93.9%) |
| Mild (1–2) | 84 (5.6%) | 1,407 (94.4%) | 111 (7.4%) | 1,380 (92.6%) | 148 (9.9%) | 1,343 (90.1%) |
| Moderate (3–4) | 46 (6.8%) | 633 (93.2%) | 65 (9.6%) | 614 (90.4%) | 92 (13.5%) | 587 (86.5%) |
| Severe (≥5) | 51 (11.3%) | 399 (88.7%) | 65 (14.4%) | 385 (85.6%) | 88 (19.6%) | 362 (80.4%) |
| **Region** | | | | | | |
| Atlantic | 16 (6.2%) | 240 (93.8%) | 24 (9.4%) | 232 (90.6%) | 31 (12.1%) | 225 (87.9%) |
| Bogota | 71 (5.2%) | 1,282 (94.8%) | 98 (7.2%) | 1,255 (92.8%) | 140 (10.3%) | 1,213 (89.7%) |
| Central | 61 (5.8%) | 990 (94.2%) | 80 (7.6%) | 971 (92.4%) | 104 (9.9%) | 947 (90.1%) |
| Eastern | 24 (10.0%) | 215 (90.0%) | 26 (10.9%) | 213 (89.1%) | 33 (13.8%) | 206 (86.2%) |
| Pacific | 37 (6.0%) | 581 (94.0%) | 52 (8.4%) | 566 (91.6%) | 75 (12.1%) | 543 (87.9%) |
| **Weight of procedure** | | | | | | |
| Isolated valve procedure | 131 (4.9%) | 2,568 (95.1%) | 175 (6.5%) | 2,524 (93.5%) | 250 (9.3%) | 2,449 (90.7%) |
| Double valve procedure | 6 (9.4%) | 58 (90.6%) | 7 (10.9%) | 57 (89.1%) | 9 (14.1%) | 55 (85.9%) |
| Isolated valve + 1 procedure | 64 (9.4%) | 616 (90.6%) | 87 (12.8%) | 593 (87.2%) | 113 (16.6%) | 567 (83.4%) |
| Double valve + ≥2 procedures | 2 (11.8%) | 15 (88.2%) | 3 (17.6%) | 14 (82.4%) | 3 (17.6%) | 14 (82.4%) |
| Isolated valve + ≥2 procedures | 6 (10.5%) | 51 (89.5%) | 8 (14.0%) | 49 (86.0%) | 8 (14.0%) | 49 (86.0%) |
| **Technique** | | | | | | |
| Surgical | 187 (6.1%) | 2,895 (93.9%) | 252 (8.2%) | 2,830 (91.8%) | 329 (10.7%) | 2,753 (89.3%) |
| Transcatheter | 18 (5.8%) | 290 (94.2%) | 24 (7.8%) | 284 (92.2%) | 47 (15.3%) | 261 (84.7%) |
| Minimally Invasive | 4 (3.1%) | 123 (96.9%) | 4 (3.1%) | 123 (96.9%) | 7 (5.5%) | 120 (94.5%) |
| **Exposure** | | | | | | |
| Teaching Hospital | 29 (2.8%) | 1,022 (97.2%) | 46 (4.4%) | 1,005 (95.6%) | 82 (7.8%) | 969 (92.2%) |
| Non-Teaching Hospital | 180 (7.3%) | 2,286 (92.7%) | 234 (9.5%) | 2,232 (90.5%) | 301 (12.2%) | 2,165 (87.8%) |

CCI: Charlson Comorbidity Index.

for the entire cohort (95% CI 6.1–7.2), 7.1 (95% CI 6.4–7.8) for those operated in a non-teaching hospital, and 5.6 (95% CI 4.7–6.6) for those operated in a teaching hospital. Further, the median age of the patients who died at 30 days was higher (68.7 (25.6–87.5) vs. 64.7 (18.1–91.5); p<0.001), the same occurring at 90 days (68.9 (25.6–87.8) vs. 64.6 (18.1–91.5); p<0.001) and one year (69.2 (23.8–88.8) vs. 64.4 (18.1–91.5); p<0.001). Detailed information on mortality per teaching hospital status and mortality per index procedure can be found in S3 and S4 Tables.

Of the entire sample, at 90 days, 828 patients (23.5%) had at least one readmission. Of all reinterventions, only a slight difference was observed when comparing the proportions by teaching hospital status (n = 46, 4.4% for teaching hospitals vs. n = 81, 3.3% for non-teaching hospitals); p = 0.112). Similarly, 127 patients (3.6%) required at least one reintervention during the first year (S5 Table).

The median age of readmitted patients was higher (65.7 (18.7–91.4) vs. 64.8 (18.1–91.5); p = 0.038), yet it was lower for those with reinterventions (62.9 (25.6–87.2) vs. 65.1 (18.1–91.5); p = 0.117). The most frequent valve reinterventions were surgical aortic valve replacement (SAVR) (n = 73, 3.5% of all surgically replaced aortic valves) and surgical mitral valve replacement (SMVR) (n = 34, 3.6% of all surgically replaced aortic valves), followed with $\leq$ six cases of other procedures.

### Inverse probability of treatment weighting (IPTW)

The standardized difference in the original sample was greater than 10% in four variables (intervention technique = 28.4%, year of intervention = 20.1%, EPS = 124%, and region = 121%). The regression model for calculating the propensity scores reports a c statistic of 0.91; however, the lack of overlap in the propensity scores between the exposed and unexposed and some extreme scores indicated that the positivity assumption is not met (i.e., there are subjects who lack at least some chance to receive any of the two exposures).

Unrestricted weights were calculated, having a 75th percentile of 1.64 (Min: 1, Max: 504.91); yet, after applying the unrestricted weights to our sample, the largest standardized difference was 27% for the variable "age group." All the above reveals the need for another alternative functional form to calculate the weights. Consequently, restricted weights were calculated, limiting the propensity scores to those in the range of 0.1–0.9, and delivered the 75th percentile of 2.11 (Min: 1.11, Max: 9.95). IPTW with restriction provided a representation of 1,889.5 and 1,935.6 patients for the teaching and non-teaching hospital groups, respectively; the largest standardized difference was 7.6% (variable "EPS"), with no values greater than 10%. This diagnosis suggests that the IPTW with restriction allows us to create a weighted artificial balanced sample (sample characteristics before and after IPTW are provided in S6 Table).

**Survival analysis and logistic regression models.** Fig 1 provides the Kaplan-Meier survival curve for both patients operated on in teaching vs. non-teaching hospitals, distinguishing the behavior of the original and the weighted sample. In contrast, we found no statistically significant differences in the weighted sample. After fitting a Cox proportional hazards model based on the weighted sample, there was no effect of being operated on in a teaching hospital on survival in terms of time-to-event of death (HR 1.07; 95% CI 0.78–1.46, p-value = 0.668). The proportional hazard assumption was verified using the Schoenfeld residuals test (p-value = 0.307). Further, five logistic regression models were fitted to the weighted sample to identify the marginal effect of exposure on our secondary outcomes (Table 3).

### Discussion

We found no effect of teaching hospital status on survival in terms of time-to-event of death in patients who underwent their first valve replacement procedure affiliated with the

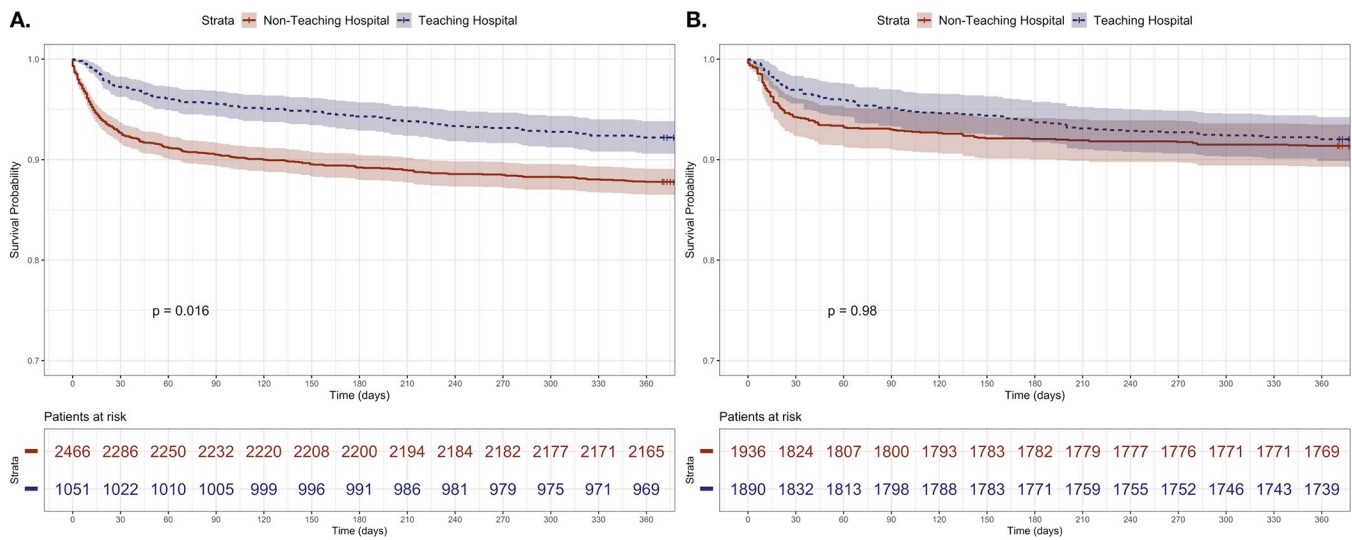

**Fig 1.** Kaplan-Meier survival curve for exposed and unexposed patients in the original (A) and weighted (B) samples.

contributory regime in Colombia for 2016–2019. As a positive finding, patients operated in a teaching hospital have a lower 30-day mortality risk than those operated in a non-teaching hospital. No effect was identified on other secondary outcomes or on time-to-death.

The 30-day mortality in our cohort (5.9%) resembled that reported by other local studies (3.9–9.2%) [19,20] and those reported internationally (2.8–7.3%) [21–23]; interestingly, it was lower than the 7.3% rate reported by the Brazilian BYPASS Registry Analysis [21]. The 90-day mortality of our cohort (8%) also seems consistent with the literature's 6% (4–8%) rate of all cardiac surgeries of 7.5–22.3% when categorized by age [24]. The one-year mortality of 10.9% is also consistent with rates reported by local studies, such as 14.3% of a nationwide population-based study in Brazil (survival of 85.7%) [25]; likely, it is in line with data from Italy (12.8%) [26]. Conversely, it is slightly higher than the 7.7% reported for Australia and New Zealand [4] or than the range from 6.2 to 9.6% reported by the STS Database [27,28]. When comparing the Latin American population with the rest of the world, these minor differences highlight the importance of understanding potential disparities due to the region's vast -ethnical, geographical, sociocultural, and political- diversity.

Interestingly, our cohort showed lower one-year mortality rates for transcatheter aortic valve replacement (TAVR) patients than those reported in the literature (15.7% vs. 17%) [29]. Furthermore, the one-year mortality for patients undergoing minimally invasive surgery was certainly low (7 cases out of 100); this encourages the need to raise awareness of such techniques in our country and region [30].

**Table 3. Logistic regression on the weighted sample for our secondary outcomes.**

|  | 30-day mortality | 90-day mortality | One-year mortality | 90-day readmission | One-year reintervention |
|---|---|---|---|---|---|
| **OR*** | 0.51 | 0.68 | 1.09 | 1.21 | 0.93 |
| **Robust SE** | 0.296 | 0.256 | 0.170 | 0.135 | 0.303 |
| **95% CI** | 0.29–0.92 | 0.41–1.12 | 0.78–1.52 | 0.92–1.58 | 0.51–1.68 |
| **p-value** | **0.024** | 0.127 | 0.627 | 0.161 | 0.812 |

OR*: Odds Ratio when considering teaching hospital status as the exposure.

Our 23.7% 90-day readmission rate appears to be consistent with the 30-day readmission rates of 13–26%, as well as with the 65-day readmission rates of 18.3–25% reported by previous studies [31–34]. In addition, one-year reintervention rates of 1–2.2% [35,36] contrast with the rate of 3.6% in our cohort. The discrepancies could be attributed to the variety of procedures and techniques included in our sample. Readmissions were more frequent in patients ≥80, with moderate-severe comorbidities who underwent a combined procedure. Reinterventions were more frequent in ≤60 female patients without comorbidities who underwent an isolated surgical procedure; we are aware that the average follow-up to assess the durability of a prosthetic valve is 10 to 15 years, so a longer follow-up time will be needed to validate any trend in our population.

Considering our exposure, mortality rates were overall higher for patients operated on in non-teaching hospitals, particularly at 30 days (p<0.001) and 90 days (p<0.001) than at one year (p<0.001); nevertheless, adjusted survival curves showed no differences. Readmissions were barely higher in patients operated in non-teaching hospitals (p = 0.464) as opposed to reinterventions (p = 0.112).

These results are consistent with those observed in earlier studies. In 1991, Sethi et al. reported for 957 valve replacements an operative mortality of 7.2% and 9.7% for those with (49.5%) and without resident assistance, respectively, without finding statistically significant differences when adjusted for specific risk factors [9]. Telila et al. in 2017, for 33,790 TAVIs in the US -89.3% performed in teaching hospitals- found no differences in adjusted mortality or serious cardiovascular events but an increase in acute kidney injury as an adverse event (OR 1.34, 95% CI 1.04–1.72), a longer length-of-stay (7.7 vs. 6.8 days) and higher average hospitalization costs (USD 50,814 vs. 48,787, p-value = 0.02) in teaching hospitals [10]. Furthermore, in 2017, Zack et al. for 5,005 tricuspid valve procedures (replacement and repair), did not observe any differences in adjusted mortality for 86% of patients operated in a teaching hospital [8].

On the other hand, we found that patients operated in a teaching hospital had a lower adjusted 30-day mortality risk. This finding broadly supports the work of other studies. In 2019, Shah et al. reported that teaching hospital status positively affected adjusted in-hospital mortality for AVR and mitral valve procedures (replacement or repair) and refuted the "July effect" (i.e., the hypothesis that the results of surgery are worse in the first month of training in which there is a new cohort of residents) for 470,005 cardiovascular surgery procedures [7]. Pant et al. in 2016 reported lower rates of in-hospital complications in teaching hospitals (42 vs. 50%; p-value <0.001) for 7,405 TAVIs in the US in 2012 (88% performed in teaching hospitals) [3]. Similarly, Gopaldas et al. in 2012 also reported lower rates of in-hospital complications in teaching hospitals for combined AVR and coronary artery bypass grafting (CABG) procedures [5].

A possible explanation for the absence of differences in our study could be that in Colombia, cardiovascular surgery programs until 2021 traditionally admitted applicants who were already specialized in general surgery and had at least 4 to 5 years of training in surgical skills. In addition, cardiovascular procedures require the highest quality of surgical instruments and postoperative surveillance standards regardless of the hospital where they are performed.

The generalizability of our results is subject to certain limitations. A more comprehensive definition of teaching hospital status is needed, so we proposed a mixed definition that includes hospitals recognized by national regulations and those providing training to cardiovascular surgery fellows. The lack of detailed clinical information (e.g., anatomical and physiological/functional complexity as well as data on the operative skills and surgical volume of each surgical team), other potential outcomes (e.g., reintervention for bleeding, paravalvular leak, and patient-prosthesis mismatch), and specific causes of readmission and mortality are inherent limitations to database studies. This is compensated by considering our sample size and

the robustness and validity that entail the use of the IPTW when controlling the effect of possible unobservable confounders through the design and assessing balance in the weighted sample of several attributes as a proxy for the preoperative and clinical complexity of the case.

Finally, one could argue that including such a diverse set of procedures and techniques would mislead the interpretation of the results. Nonetheless, the main push of our study was to provide previously unreported data enclosing a comprehensive array of major valve replacement procedures transverse to the extent of health care on a national scale; therefore, we included all valve replacements disregarding the treated valve, the weight of the procedure and the technique.

Further prospective, valve- and technique-specific studies should confirm the conclusions drawn from our work. Given the known needs, we advise the urgent creation of a national (or regional) information system for cardiovascular surgery that includes not only clinical data but also data related to the physician workforce, and we support recent efforts to build a national database led by the Colombian Society of Cardiology and Cardiovascular Surgery.

## Conclusions

After IPTW, the 30-day mortality odds are lower for those operated in a teaching hospital. There was no effect on survival, 90-day or one-year mortality, 90-day readmission, or one-year reintervention. This study encourages the use of causal inference approaches applied to administrative claims data to assess questions that would otherwise be unethical and unfeasible. Taken together, we offer an opening for investigating an exposure that has yet to be explored in Latin America with potential value not only for cardiovascular medicine but also for understanding teaching hospitals as the essential nature of reality of an academic-clinical synergy.

## Supporting information

**S1 Table. Region definitions.**
(PDF)

**S2 Table. Frequencies of valve replacement procedures and other concomitant cardiovascular surgical procedures performed simultaneously with the "index procedure" and frequencies of valve reinterventions.**
(PDF)

**S3 Table. Cumulative incidences of death at 30 days, 90 days, and one year per teaching hospital status.**
(PDF)

**S4 Table. Cumulative incidences of death at 30 days, 90 days, and one year per index procedure.**
(PDF)

**S5 Table. Cumulative incidences of 90-day readmission and one-year reintervention per teaching hospital status.**
(PDF)

**S6 Table. Baseline characteristics before and after inverse probability of treatment weighting with restriction.**
(PDF)

**S1 Text. R session.**
(PDF)

## Author Contributions

**Conceptualization:** Kevin Maldonado-Cañón, Giancarlo Buitrago, Germán Molina, Francisco Mauricio Rincón Tello, Javier Maldonado-Escalante.

**Data curation:** Kevin Maldonado-Cañón.

**Formal analysis:** Kevin Maldonado-Cañón.

**Investigation:** Kevin Maldonado-Cañón, Giancarlo Buitrago, Germán Molina, Francisco Mauricio Rincón Tello, Javier Maldonado-Escalante.

**Methodology:** Kevin Maldonado-Cañón, Giancarlo Buitrago.

**Project administration:** Kevin Maldonado-Cañón.

**Software:** Kevin Maldonado-Cañón.

**Supervision:** Giancarlo Buitrago, Germán Molina, Francisco Mauricio Rincón Tello, Javier Maldonado-Escalante.

**Writing – original draft:** Kevin Maldonado-Cañón.

**Writing – review & editing:** Kevin Maldonado-Cañón, Giancarlo Buitrago, Germán Molina, Francisco Mauricio Rincón Tello, Javier Maldonado-Escalante.

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
