## [Decision Letter · Decision Letter 0]

22 Jun 2023

PONE-D-23-15501

Teaching Hospitals and Their Influence on Survival After Valve Replacement Procedures: A Retrospective Cohort Study using the IPTW

PLOS ONE

Dear Dr. Maldonado-Cañón,

Thank you for submitting your manuscript to PLOS ONE. After careful consideration, we feel that it has merit but does not fully meet PLOS ONE’s publication criteria as it currently stands. Therefore, we invite you to submit a revised version of the manuscript that addresses the points raised during the review process.

We look forward to receiving your revised manuscript.

Kind regards,

Redoy Ranjan, MBBS, MRCSEd, Ch.M., MS (CV&TS), FACS

Academic Editor

PLOS ONE

Journal Requirements:

Reviewers' comments:

Reviewer's Responses to Questions

**Comments to the Author**

1. Is the manuscript technically sound, and do the data support the conclusions?

Reviewer #1: Yes

Reviewer #2: Partly

Reviewer #3: No

2. Has the statistical analysis been performed appropriately and rigorously? 

Reviewer #1: Yes

Reviewer #2: Yes

Reviewer #3: Yes

3. Have the authors made all data underlying the findings in their manuscript fully available?

Reviewer #1: Yes

Reviewer #2: Yes

Reviewer #3: Yes

4. Is the manuscript presented in an intelligible fashion and written in standard English?

Reviewer #1: Yes

Reviewer #2: No

Reviewer #3: No

5. Review Comments to the Author

Reviewer #1: In this paper, the authors aimed to evaluate the effect of the teaching hospital status on the survival of patients affiliated with the contributory regime in Colombia who underwent a first heart-valve replacement. They also performed an IPTW analysis to account for baseline differences between the two groups. The current literature is controversial; while this study supports the non inferiority of teaching hospitals with possible better short term outcomes in those institutions.

- The authors state that using conventional PS matching was not feasible. How many matched pairs were they able to identify and what was the between group baseline discrepancies?

- Provide data in tables per group (teaching vs non teaching)

- The quality of the figures are low. Could you please provide higher quality ones?

Reviewer #2: This is a retrospective cohort study based on administrative records in which the authors evaluated the effect of the teaching hospital status on the survival of patients who underwent a first heart-valve replacement. The data of 3517 patients underwent to invasive treatment of a heart valve disease were included in the analysis. The authors included all the intervention techniques (traditional surgery, minimally invasive surgery, transcatheter procedures), the treatment of all heart valve and the single or combined procedures. Their primary outcome was time-to-event of death; the secondary outcomes were the cumulative incidences of mortality at 30 days, at 90 days, and at one year, readmission at 90 days and reintervention at one year.

In conclusion, the results of this study show no effect of the teaching hospital status on survival in terms of time-to-event of death in patients undergoing their first valve replacement procedure. In addition, the patients operated on at a teaching hospital have a lower 30-day mortality risk than those operated on at a non-teaching hospital. No effect was identified on other secondary outcomes nor on time-to-death.

The topic of this study is very interesting.

However, there are some points of discussion:

1. The manuscript needs an English revision.

2. In the title the authors wrote “Valve Replacement Procedures” but they included in the analysis also the valve repair. It could be necessary to change the title.

3. The authors included in the analysis all the intervention techniques (traditional surgery, minimally invasive surgery, transcatheter procedures), the treatment of all heart valve and the single or combined procedures. To better understand the difference between teaching and no- teaching hospitals, it would be better to focus on a single valve procedure or on a single type of intervention technique.

4. The primary source of information was the Capitation Sufficiency Database from the 94 Colombian Ministry of Health and the patient’s data were assumed from demographic information and the associated ICD-10 codes. This has meant that much information regarding preoperative risk factors and the clinical course of patients is not available for analysis. Although the authors have included this topic in the limitations of the study, it’s an important shortcoming that should be filled.

5. In the “Discussion” section, the authors focused the discussion on comparing their mortality rates with the results in the literature. I suggest to better argue the topic of the study, namely the difference in the outcomes between the teaching and no- teaching hospitals.

Reviewer #3: The Authors submitted a Research Article set on teaching Hospitals and based on their influence on survival after valve replacement procedures through a retrospective cohort study using the inverse probability of treatment weighting (IPTW). So, they concluded that they presented a first step towards researching an exposure that has yet to be studied in our country with potential value not only for cardiovascular medicine but also for understanding the teaching hospitals as the essential nature of reality of an academic-clinical synergy.

Although the manuscript is well written, it must be clearer. I need help understanding the aims and the scope of this study. The title should be rephrased; IPTW should be elucidated. The Aims of the study should be declared and uniformed in the background of the abstract and the Introduction's end. The conclusions should present what the study really found and not stating that "present a first step..." and phrases like that.

Finally, in the "Background", both abstract and text, it should be clarified why IPTW is essential.

6. PLOS authors have the option to publish the peer review history of their article (what does this mean?). If published, this will include your full peer review and any attached files.

Reviewer #1: No

Reviewer #2: No

Reviewer #3: **Yes: **Francesco Bianco

---

## [Author Response · Author response to Decision Letter 0]

3 Aug 2023

We thank the reviewers for a thorough and careful reading of our work and for the thoughtful comments and constructive suggestions that helped improve the quality of this manuscript.

Please find attached the "Response to Reviewers.pdf" letter, where we respond to individual remarks point-by-point.

---

## [Decision Letter · Decision Letter 1]

15 Aug 2023

Teaching hospitals and their influence on survival after valve replacement procedures: A retrospective cohort study using inverse probability of treatment weighting (IPTW)

PONE-D-23-15501R1

Dear Dr. Maldonado-Cañón,

We’re pleased to inform you that your manuscript has been judged scientifically suitable for publication and will be formally accepted for publication once it meets all outstanding technical requirements.

Kind regards,

Redoy Ranjan, MBBS, MRCSEd, Ch.M., MS (CV&TS), FACS

Academic Editor

PLOS ONE

Additional Editor Comments (optional):

Review Comments to the Author

Reviewer #1: Thank you for addressing my comments . There are no further comments from my side. Thank you foe the opportunity to review this paper

Reviewer #3: The Authors addressed all the comments raised from the Reviewers, therefore, the manuscript significantly improved after this revision. I have no further comments or edits.

---

## [Editor Report · Acceptance letter]

17 Aug 2023

PONE-D-23-15501R1 

Teaching hospitals and their influence on survival after valve replacement procedures: A retrospective cohort study using inverse probability of treatment weighting (IPTW) 

Dear Dr. Maldonado-Cañón:

I'm pleased to inform you that your manuscript has been deemed suitable for publication in PLOS ONE. Congratulations! Your manuscript is now with our production department. 

Kind regards, 

on behalf of

Dr. Redoy Ranjan 

Academic Editor

PLOS ONE